# Cancer Stem Cells in Tumours of the Central Nervous System in Children: A Comprehensive Review

**DOI:** 10.3390/cancers15123154

**Published:** 2023-06-11

**Authors:** Yi-Peng Han, Hou-Wei Lin, Hao Li

**Affiliations:** 1Department of Neurosurgery, Children’s Hospital of Fudan University, National Children’s Medical Center, Shanghai 201102, China; yi.peng.han.nagoya@gmail.com; 2Department of Paediatric Urology, Xinhua Hospital Affiliated to Shanghai Jiao Tong University School of Medicine, Shanghai 200092, China; 3Department of Paediatric Surgery, Jiaxing Women and Children Hospital Affiliated to Jiaxing University, Jiaxing 314001, China

**Keywords:** cancer stem cells (CSCs), tumours of the central nervous system (CNS), children, biology impact, clinical significance

## Abstract

**Simple Summary:**

Tumours of the central nervous system (CNS) are the most common solid tumours in children. The difficulties in treating this complicated variety of tumours are that they lead to high mortalities. Cancer stem cells (CSCs) are a subgroup of cells found in various kinds of tumours with stem cell characteristics, such as self-renewal, induced differentiation, and tumourigenicity. The biological functions and clinical significance of CSCs in CNS tumours in children have been reported in the last few decades. However, there is no comprehensive review in this field and the information has been fragmented and confusing. Thus, we collected valuable data and findings from these reports to make a clear and concentrated review to provide a train of thought for experts in the field.

**Abstract:**

Cancer stem cells (CSCs) are a subgroup of cells found in various kinds of tumours with stem cell characteristics, such as self-renewal, induced differentiation, and tumourigenicity. The existence of CSCs is regarded as a major source of tumour recurrence, metastasis, and resistance to conventional chemotherapy and radiation treatment. Tumours of the central nervous system (CNS) are the most common solid tumours in children, which have many different types including highly malignant embryonal tumours and midline gliomas, and low-grade gliomas with favourable prognoses. Stem cells from the CNS tumours have been largely found and reported by researchers in the last decade and their roles in tumour biology have been deeply studied. However, the cross-talk of CSCs among different CNS tumour types and their clinical impacts have been rarely discussed. This article comprehensively reviews the achievements in research on CSCs in paediatric CNS tumours. Biological functions, diagnostic values, and therapeutic perspectives are reviewed in detail. Further investigations into CSCs are warranted to improve the clinical practice in treating children with CNS tumours.

## 1. Introduction

Cancer is one of the largest health problems worldwide and is one of the leading causes of death in the 21st Century [1]. Tumours of the central nervous system (CNS) rank fourteenth by incidence in all kinds of cancers in both males and females worldwide [1], and are among the top ten cancer mortalities in spite of gender [2,3]. In children, CNS tumours are the most common solid tumours, with an age-standardized incidence rate per million person-years (ASR) of 28.2, accounting for 17.2–26.3% of paediatric malignancy [4]. After being deeply studied, revolutionary molecule-based classifications in the new edition of the WHO Classification of Tumours of the Central Nervous System were introduced [5]. However, only a few novel treatments were introduced, such as BRAF-related targeting therapies in low-grade gliomas, and preliminary studies of histone deacetylase (HDAC) inhibitors in paediatric high-grade gliomas [6], necessitating the development of new therapeutic strategies.

Cancer stem cells (CSCs) were first reported in leukaemia in the late 20th Century, which played an important role in leukaemogenesis, as tumour initiation cells [7]. Although CSCs were later identified in many types of solid tumours and haematological malignancies, the heterogeneous nature of various malignancies, as well as phenotypical differences among patients with the same cancer type, mitigated the efforts to identify, understand, and develop targeted therapies against CSCs [8,9]. Tumour stem-like cells were pioneers in anaplastic astrocytoma and glioblastoma tissue in 2002 [10], and identified in several kinds of paediatric CNS tumours, such as pilocytic astrocytoma, medulloblastoma, ganglioglioma, anaplastic astrocytoma, glioblastoma multiforme, and ependymoma afterward [11,12]. Neurosphere assay is a standard procedure in insolating neural stem cells and deriving CNS CSCs, suggesting that CSCs in CNS were transformations of undifferentiated neural precursor cells [13]. Here, we review the recently published reports on basic research advances and clinical implications of CSCs in CNS tumours in children to shed light on the importance of this potential therapeutic target.

## 2. Tumour Stem Cells in Major Types of Tumours of the Central Nervous System in Children

According to the new edition of the WHO Classification of Tumours of the Central Nervous System, tumours of the CNS in children are graded and classified according to a combination of histological and molecular characteristics, the clinical features of which and pathways of tumourigenesis are significantly different from adult tumours [14]. Here, we review the origins and markers of CSCs in the following major types of tumours of the CNS in children (Table 1).

### 2.1. Gliomas, Glioneuronal Tumours, and Neuronal Tumours

The catalogue of gliomas, glioneuronal tumours, and neuronal tumours in the 2021 WHO classification contained unaltered types from the 2017 edition, such as ependymal tumours, and glioneuronal and neuronal tumours, while their subtypes expanded by divergent molecular signatures. Paediatric-type diffuse low-grade gliomas and paediatric-type diffuse high-grade gliomas were new types of tumours to clearly separate these prognostically and biologically distinct groups of tumours from adult ones, as well as circumscribed astrocytic gliomas that substituted other astrocytic tumours in the 2017 WHO classification [5,62].

The most common low-grade glioma in children was pilocytic astrocytoma, belonging to the tumour type of circumscribed astrocytic gliomas, caused by extracellular-signal-regulated kinase (ERK) constitutive activation from its upstream BRAF mutations (BRAF-KIAA1549 fusion or BRAF-V600E) or NF1 alterations [63], and the less common active mutations of FGFR1 and PTPN11, and NTRK2 fusion [64]. The CSCs from pilocytic astrocytoma were identified in the highly pioneering work in insolating CSCs from CNS tumours by primary tumour sphere culture [11]. CD133 and Nestin were first used as CSC markers for pilocytic astrocytoma, while pluripotent markers SOX2, Nanog, Oct4, CD44, Integrin α6 (CD49f), SSEA-1, or CD15, ATP Binding ABCG1, and neuro-stem cell markers such as Podoplanin, and markers for different precursor/progenitor cells such as BLBP and A2B5, were later introduced by different research groups [11,23,24,25,26]. Specific markers to confirm the differentiation status of pilocytic astrocytoma CSCs were reported as Beta-III Tubulin, GFAP, Oligodendrocyte Marker O4, S100B, NF, and EAAT1/2 [11,24,25,26]. There were also controversial markers such as Olig2 and PDGFRα, which were regarded as not only CSC markers, but also differentiation markers, in different reports [24,26,27,28]. This situation was ascribed to the definition of CSCs in CNS, since neuro-stem cells, neural precursor cells, and even neuron/oligodendrocyte/glial progenitor cells were all transformable to the CSCs in generating the tumours in different studies [13]. The latest hypothesis was that pilocytic astrocytoma CSCs were more differentiated radial glia/oligodendrocyte precursor cell-like cells than those immature neuro stem cell-like CSCs in high-grade gliomas [28]. In pleomorphic xanthoastrocytoma, another low-grade circumscribed astrocytic glioma, CSCs were preliminarily studied and it was revealed that CD133 was low, while a high level of CD15 was detected, with gradually decreased GFAP-positive cells by passages, in a xenograft model [29]. In other low-grade gliomas, CSCs were expectedly discovered by the recognition of CSC markers of CD133, SOX2, Nestin, and the confirmation of their differentiation by oligodendrocyte lineage markers of CNPase, glial markers of GFAP, and NFP, in oligodendroglioma (defined as diffuse low-grade glioma, MAPK pathway-altered in the 2021 WHO classification) [15]; by CSC markers of SOX2, AGR2, and differentiation markers of GFAP, Beta-III Tubulin, in dysembryoplastic neuroepithelial tumours [30]; by CSC markers of CD133, Nestin, SOX2, Nanog, CD44, EGFR, hTERT, BLBP, GFAP-delta, Olig2, and ASCL1; and by differentiation markers of Beta-III Tubulin, GFAP, Synaptophysin, MAP2, NeuN, BMP2, BMPR1B, and PSANACM in central neurocytoma [31,32].

High-grade gliomas in children are relatively less common, but their prognosis is extremely poor. Since the WHO 2021 classification withdrew glioblastoma in children and established paediatric-type diffuse high-grade gliomas as a new catalogue, old entities of glioblastoma, anaplastic astrocytoma, diffuse intrinsic pontin gliomas (DIPG), and other high-grade gliomas were reclassified to the new tumour type. The CSCs of paediatric high-grade gliomas were isolated as early as 2003, with stemness markers of Nestin and Musashi, along with the differentiation markers of Beta-III Tubulin, GFAP, and Oligodendrocyte Marker O4 [12]. Later, CD133, SOX2, CD44, SEEA-1, ALDH, L1CAM, Olig2, Nanog (partial, case-by-case), Podoplanin, and Bmi-1 were recognized as additional CSC markers, and MAP2, NFP, CNPase, NeuN, and MBP were introduced as differentiation markers [15,20,21,22]. Additionally, in H3 K27-altered Diffuse Midline Gliomas (DMG), a group of devastating highly malignant tumours raised in children, the driver mutation itself could have interfered with differentiation and promoted stem cell proliferation with maintained stemness [65]. Studies on DIPGs by human-derived primary cell lines or iPSC harbouring H3K27M not only revealed consistent CSC markers of CD133, SOX2, Nestin, PAX6, Vimentin, and differentiation markers of MAP2, GFAP, NeuN, NOTCH 1, and CSPG4 in previous studies on high-grade gliomas, but also introduced new CSC markers of ALDH and controversial markers of Olig2 and PDGFRα; due to the origins of DIPG, CSCs differed in different research groups [16,17,18,19].

Ependymal tumours are groups of tumours derived from the glial cell lining of the ventricular system, the subgroups of which have been carefully sorted by molecular signatures and anatomic locations. These tumours can develop in both children and adults, and treatments are challenging due to the locations and chemo-radiation resistances of tumours in children. Although the driven mutations of the supratentorial ependymoma (ZFTA fusion-positive or YAP1 fusion-positive) and posterior fossa ependymoma (either group PFA or PFB) are fully elucidated, according to the WHO 2021 classification [5,66], there is still no efficient treatment to control them. In the ground-breaking work from 2005, CSCs of ependymoma were supposed to be a transformation of radial glia cells (RGCs), since ependymomas recapitulated the gene expression profiles of regionally specified RGCs. The CSC markers of Nestin, CD133, RC2, BLBP, and differentiation markers of β-III Tubulin, MAP2, GFAP, S100, CNPase, and NG2 were accepted [33]. In later studies, the driver mutations of ZFTA (RELA) fusion and YAP1 fusion were both confirmed to contribute to the oncogenic signalling by inducing neural progenitor cells or neural stem cells to form ependymomas [67,68], and single-cell RNA sequencing revealed subgroups of tumour cells called undifferentiated ependymal cells (UECs) that might act as CSCs in ependymomas, since their RNA profile overlapped with most of progenitors in different lineages, with spatiotemporally specific signatures in separating CSCs of supratentorial and anteroposterior ependymoma [35,36]. In total, the CSC markers for ependymoma were expanded (there were numerous markers from single-cell sequencing and, here, we list important ones) to include Vimentin, HES1, PBX1, SOX9, PAX3, Protein c-Fos, ZFP36, LGR5, EGR1, JUN, and ATF3, and differentiation markers of Synaptophysin, O4, Olig1/2, APC, CSPG2, DNAAF1, RSPH1, and CAPS were added [34,35,36].

### 2.2. Choroid Plexus Tumours

Choroid plexus tumours originate from choroid plexus epithelial cells, which derive from neuroepithelial progenitors with MYC overexpression and a loss of p53, and are extremely rare in adults but are commonly seen in young children under 1 year of age [37,38]. According to the 2021 WHO classification, three subtypes of choroid plexus tumours were grouped as choroid plexus papilloma, atypical choroid plexus papilloma, and choroid plexus carcinoma [5]. Although recent molecular subgroups based on epigenetic profiles have been matched with subtypes in view of pathological characteristics, patients with choroid plexus papilloma still have poor prognoses, with a 5-year overall survival of 65% [69,70]. Since choroid plexus papilloma is regarded as a fully differentiated benign papillary neoplasm closely resembling non-neoplastic choroid plexus tissue, most studies on CSCs of choroid plexus tumours have focused on choroid plexus carcinoma [38]. The markers for CSCs in choroid plexus carcinoma are MYC, Nestin, ATOH1, BLBP, GFAP, Geminin, GDF7, and differentiation markers are mainly for choroid plexus epithelial cells, such as TTR, AQP1, OTX2, GMNC, MCIDAS, FOXJ1, CCNO, TAp73, and MYB [37,38,39].

### 2.3. Embryonal Tumours

CNS embryonal tumours are different groups of highly malignant tumours mainly affecting young children, with gradually increased incidence over a long time period [6,71]. In the new WHO classification, the two types of embryonal tumours are medulloblastomas and other CNS embryonal tumours. Molecular subtypes of medulloblastomas are extraordinarily famous and practical due to their perfect correlation with clinical features [5]. The first report of CSCs in medulloblastoma was the pioneering work on isolating multiple types of paediatric brain tumours in 2003, with anaplastic astrocytoma and glioblastoma, by the same CSC markers as Nestin and Musashi [12]. The mouse model confirmed that the SHH subgroup medulloblastoma arose from both granule neuron precursors (GNPs) and multipotent neural stem cells, by CSC markers of p75NTR, TrkC, Zic1, MATH1, SOX1, SOX2, PLZF, DACH1, Multimerin 1, PAX6, ATOH1, MycN, and differentiation markers of NeuN, GABRA6, Synaptophysin, PAX2, BLBP, and O4 [40,41]. Then, in medulloblastoma cell lines of SHH, group3/4 subtypes, CSCs were successfully discriminated by markers of CD133, Nestin, SOX1, and SOX2, and markers of GFAP, CD44, CD24, and Beta-III Tubulin were applied to indicate differentiation [42]. Furthermore, an interesting phenomenon had been noticed that, in all four subgroups of medulloblastoma, a small group of Wnt-active cells existed and could impair the stemness of CSCs by reducing Bmi-1 and SOX2 levels [72].

Embryonal tumour with multilayered rosettes (ETMR) is another type of common aggressive embryonal tumour in young children, with C19MC-altered or DICER1-altered molecular signatures. Although ETMRs consist of populations resembling neural stem cells, radial glial cells, and more differentiated cells, the neuronal and glial cell markers can be detected in limited parts of tumour tissues [48]. CSC components have been found in different histological subtypes of ETMRs by markers of LIN28A/B, HMGA2, Nestin, Vimentin, Oct4, SOX2, Nanog, CRABP1, DNMT3B, SOX3, SOX11, PAX6, SALL4, POU3F2, MEIS1/2, MYCN, Wee1, and CHEK2, and their induced differentiation cells have been marked with AQP4, GFAP, Synaptophysin, NeuN, and NFP [47,48,49].

Atypical teratoid/rhabdoid tumours (AT/RT) have been divided into three molecular subtypes, TYR, SHH, and MYC, which predominantly affect infants or young children, with a remarkably simple alteration of SWItch/sucrose nonfermentable related, Matrix-associated, Actin-dependent regulator of Chromatin, Subfamily B1 (SMARCB1) or SMARCA4, but with devastating outcomes [73]. After the first report of isolating CSCs of AT/RT from patients by a marker of CD133 through sphere culture and a xenograft model [43], SMARCB1-deficient pluripotent stem cells-based experiments revealed that the origin of CSCs in AT/RT, SMARCB1-deficient neural progenitor-like cells efficiently gave rise to AT/RT-like tumours and their stemness signatures worsened the prognosis [46]. The markers of AT/RT CSCs are general markers of CD133, Nestin, Musashi, Nanog, Oct4, and SOX2, with ALDH, SALL4, MYC, LIN28A/B, NCAM, PAX6, and KLF4, and the induced differentiation markers are MAP2, Vimentin, GFAP, synaptophysin, CD99, S-100, EMA, and SMA [43,44,45,46].

### 2.4. Germ Cell Tumours

Intracranial germ cell tumours (GCT) are less seen in children in Western countries as they account for 0.4–3.4% of all paediatric CNS tumours, while in Asia, they account for as high as 11%. Since GCTs have various types based on their cell components, the origins of GCTs have been explored by experts with long-standing controversy in “germ cell” and “pluripotent stem cell” theories [74]. According to early reports of GCTs, stem-cell-related proteins of C-KIT, Oct3/4, transcription factor AP-2 gamma (TFAP2C, or AP-2γ), Nanog, and germ-cell-specific proteins of Melanoma-associated antigen A4 (MAGE-A4), cancer-testis antigen 1B (CTAG1B, or NY-ESO-1), expressed in tumour tissue, and primordial germ cells are regarded as tumour generators to convince “germ cell” theory [75]. Nevertheless, the Oct4-activated neural stem cell can trigger the formation of teratoma in the brain and raise the possibility of “pluripotent stem cell” theory [76]. However, it has been widely accepted that germinoma is a prototype of all GCTs, by CSC markers of Oct4, Nanog, SOX17 (germinoma), SOX2 (infantile GCTs), and differentiation markers of PLAP and KIT (germinoma differentiation markers from ES cells, rare in intracranial none germinoma GCTs (NGGCTs)) [50]. Due to the complexity of tumour composition, it is inaccurate to use markers or histopathology alone using small specimens for diagnosis, also in the identification of CSCs in GCTs [77]. Recent milestone research has supposed that germinomas are raised from primordial germ cells, while NGGCTs originate from embryonic stem cells by transcriptome and methylome analysis, and CSC markers are separated into different subtypes, such as Lin28A, SOX2 KLF2/4 as general CSC markers, PIWIL1, DAZL, DDX4, NANOS3, and ERVW-1 for germinomas, and Nanog and Oct4 for NGGCTs [51].

### 2.5. Tumours of the Sellar Region

The most seen tumour in the sellar region in children is the adamantinomatous craniopharyngioma (ACP), which is a distinct type of tumour other than papillary craniopharyngioma in the WHO 2021 classification, due to their different clinical demographics, radiologic features, histopathologic findings, genetic alterations, and methylation profiles [5]. Interestingly, several studies found that CSCs of ACPs shared the same origin with paediatric rare tumour of pituitary adenoma, from pituitary stem cells (PSCs) [53]. The CSC markers for ACPs were Nestin, SOX2, Oct4, CD133, KLF4, SOX9, β-catenin, MYC, SCA1, HESX1, and KLF2/4, and the differentiation markers were similar to pituitary hormone-secreting cells, such as PIT-1 (or POU1F1), TBX19 (or TPIT), and SF-1 (or NR5A1) [52,53,54]

### 2.6. Other Tumours

In other less common assorted CNS tumours in children, CSCs differ from each other due to their different oncogenesis alterations and tumour origins.

In a catalogue of pineal tumours, pineoblastoma caused by the recurrent homozygous deletion of DROSHA and the microduplication of PDE4DIP in primitive neuroepithelia acted as an aggressive malignancy, and their CSCs were marked with CD133, Musashi, Podoplanin, and neuro-glial differentiation by Beta-III Tubulin [55,56].

In cranial and paraspinal nerve tumours, neurofibroma and its high-grade form malignant peripheral nerve sheath tumour (MPNST), mainly presented in the department of neurology or dermatology, were more frequently seen than schwannoma in children. CSCs in these tumours were somewhat different, the markers of which were PLP, Nestin, P75, GAP43, and Sox10, with GFAP, S100, and GAP43 (as a Schwann cell marker) in differentiated cells, in neurofibroma and MPNST [58,59]; and Oct4, SOX2, Nanog, MYC, KLF4, CD133, CD44, and CXCR4 in CSCs of schwannoma [57].

CNS Ewing sarcoma belongs to the catalogue of mesenchymal, non-meningothelial tumours, driven by EWS gene fusions, which might originate from mesenchymal stem cells since their CSCs are confirmed by CD44, CD59, CD73, CD29, and CD54, CD90, CD105, and CD166, with differentiation markers of chondrocyte lineage-SOX9, COL10A1, PPARg2, FABP4, LPL, and osteogenic differentiation SPP1, ALPL, and RUNX2 [60].

Meningioma is rare in young children and its CSCs have been mainly reported and isolated from adult patients, with markers of Oct4, SOX2, Nanog, MYC, KLF4, CD133, and Nestin [61].

## 3. Major Biological Impact of Tumour Stem Cells in Tumours of the Central Nervous System in Children

The biological functions of CSCs contribute to all major aspects of tumourigenesis, tumour progression, resistance in anti-tumour therapies, and tumour relapse. Since CSCs have been named due to their stem cell characteristics, their abilities of expansion and differentiation into different lineages through self-renewal and pluripotency are self-evident [78]. In CSCs from the tumours of CNS, the origins of tumours differ as embryonic stem cells, neural crest cells, neural stem cells, neural progenitor cells, and oligodendrocyte-progenitor cells according to the neurodevelopmental processes, and tumour types, grades, and malignancies have been tightly associated to the origins of CSCs or the components of CSC subgroups [79,80]. Nevertheless, their roles in tumours depend on the following biological functions: (1) CSCs secrete angiogenic factors or even transdifferentiate into vascular endothelial cells in promoting tumour angiogenesis; (2) CSCs are highly metabolically adapted to help tumours survive the low oxygen, acidic pH, and low nutrient availability in the hypoxic niche; (3) CSCs are enriched in their capacity for both immune evasion and immunosuppression (stemness signatures are negatively correlated with PD-L1 expression); (4) CSCs secrete a variety of cytokines or ligands to transform normal fibroblasts into cancer-associated fibroblasts in supporting CSCs; (5) CSCs upregulate the epithelial-mesenchymal transition (EMT) signalling and change in adhesion receptor expression in helping tumour invasion and migration, and even convert non-CSCs into CSCs, in promoting metastasis; (6) CSCs express multidrug resistance proteins, such as ATP-Binding Cassette (ABC) transporters, protecting tumours from chemotherapy [81,82,83]. Major biological functions are illustrated with main markers in CSCs from selected types of paediatric CNS tumours (Figure 1).

### 3.1. CSCs in Neovascularization

A microenvironment with adequate oxygen and nutrition is essential for tumourigenesis and maintenance. The perivascular niche might be the best tumour ecosystem for CSCs to conduct this role, and it has been perhaps best characterized in glioblastoma, since an early study revealed that glioblastoma CSCs intimately contacted with the aberrant tumour vasculature and maintained self-renewal by several factors through autocrine and paracrine in this special environment [84]. Afterward, in CSN tumours other than glioblastoma, such as medulloblastomas, ependymomas, and oligodendrogliomas, it was confirmed that CSCs, marked with CD133 and Nestin, were closely located next to capillaries [85].

There are several ways CSCs function in core vascularization; the first and most easily accessible path is the secretion of vascular factors by CSCs to promote both angiogenesis and vasculogenesis in peritumoural vessels. It was reported in 2006 that VEGF was highly expressed by CD133-positive glioblastoma CSCs [86]. Later, in ependymoma and medulloblastoma, similar phenomena were observed in that VEGF, dickkopf WNT signalling pathway inhibitor 3 (DKK3), EGF, FGF, and PDGF from CSCs intensively enhanced neovascularization and correlated with patient outcomes [87,88,89,90]. Notably, although fewer but wider vessels and a relatively lower turnover of endothelial and tumour cells were found in pilocytic astrocytoma, in comparison with glioblastoma, the critical overlap in vessel immaturity/instability and the angiogenic profile was seen between both tumours [91].

Another path was the transdifferentiation of CSCs, which was groundbreakingly reported in glioblastoma, from an accidental discovery of p53 alterations in endothelial cells of glioblastoma capillaries, indicating that a significant portion of the vascular endothelium has a neoplastic origin [92]. The hypoxia condition, with additional VEGF, induced glioblastoma CSCs differentiated with expression patterns of vascular endothelial cell markers [93]. This transdifferentiation performance was also confirmed in Oct4-positive CSCs in neuroblastomas through VEGF stimulation [94]. Although there are many signalling pathways involved in the CSC-endothelium transdifferentiation path, reports from the tumours of CNS in children are still limited. The presentation of this phenomenon in post-chemo radiation glioblastoma has alerted the urgency for further understanding and overcoming this crucial process [95,96].

### 3.2. CSCs in the Hypoxic Niche

Hypoxia acts as an extremely important role in CSC maintenance, since it can induce other important roles of CSCs, the angiogenesis, immunosuppression, and EMT [82,97,98]. The high metabolism caused by the proliferation and migration of cancer expands the necrosis in tumour tissue, commonly seen in highly malignant tumours. These necrotic areas with hypoxic and acidic niches increase CSC maintenance and therapeutic resistance. In the early exploration of medulloblastoma, it was recognized that hypoxia-induced hypoxia-inducible factor (HIF)-1α could sustain Neurogenic locus notch homolog protein 1 (Notch 1) in its active form by preserving medulloblastoma CSC viability and expansion [99]. It has been recently revealed in ependymoma that PFA ependymomas are initiated from a cell lineage that resides in restricted oxygen [100]. The mechanism of this function was later extensively investigated, and it was clear that HIF1a/STAT3 co-activator complex induced Vasorin expression to reduce Notch turnover, augmenting Notch signalling under hypoxic stress in CSCs from glioblastomas [101]. In glioblastoma, low-pH conditions displayed a consistent increase in cancer stem cell markers, including Olig2, Oct4, and Nanog, and increased production of VEGF in promoting angiogenesis [102].

In the hypoxic niche, the function of nutrition uptake also reorientates in the CSCs of glioblastoma, in that they outcompete for glucose uptake by co-opting the high-affinity neuronal glucose transporters, which correlate with pluripotency in glioblastoma CSCs and the outcomes of patients [103].

The new technology of the organoid culture system has made it possible to investigate the hypoxic niches that were difficult to mimic through normal culture methods in finding a rapidly dividing outer region of CSCs surrounding a hypoxic core of primarily non-stem senescent cells and diffuse, quiescent CSCs in glioblastoma [104]. However, the devastating discovery of dedifferentiation in glioblastoma cells under hypoxic conditions through regulators of HIF and SOX2 [105] has made researchers and oncologists reconsider the position of the hypoxic niche in treatments targeting CSCs.

### 3.3. CSCs in Immune Evasion and Immunosuppression

Immune evasion and immunosuppression are basic abilities of cancers to survive in immunocompetent hosts. In the classical theory of tumour immunology, tumour immune evasion and immunosuppression contain several key elements, including defective tumour antigen presentation, derived immunosuppressive regulatory cells, the production of immunosuppressive mediators, dysregulated costimulatory molecules expression, immune deviation, and induce apoptosis of tumour-specific immune cells [106]. CSCs follow these procedures and enhance function in immune evasion and immunosuppression [107].

It had been confirmed that both glioblastoma CSCs and paired mature glioblastoma tumour cells are weakly positive and negative for MHC-I, MHC-II, and NKG2D ligand molecules, with defective antigen-processing machinery molecules [108]. Evidence has shown that CSCs induced Treg cells through TGF-β1 and Treg chemokine attractant CCL-2 [109], and immunosuppressive macrophages/microglia through soluble colony-stimulating factor, macrophages inhibitory cytokine-1 [110], to inhibit T-cell proliferation. The secretion of immunosuppressive mediator Prostaglandin E2 (PGE2) induced by irradiation from glioblastoma CSCs even enhanced cell survival and proliferation, causing tumour recurrence afterward [111], and macrophage migration inhibitory factor increased the production of the immune-suppressive enzyme arginase-1 and stimulated myeloid-derived suppressor cell function in glioblastoma [112]. Programmed cell death ligand 1 (PD-L1) is a member of the B7 family of co-stimulatory/inhibitory molecules, which perhaps is one of the most famous targets of the last decade in the anticancer field. Highly expressed PD-L1 in glioblastoma CSCs was recovered in the mechanism of the EMT/β-catenin/STT3/PD-L1 signalling axis [113]; however, it was still difficult to treat patients with glioblastoma, as the clinical trials are yet to reveal significant improvement in terms of outcomes [114]. This could be blamed not only on the brain-blood barrier, but also the unaffected treatment, and the PD-L1 independent PD-1 function in glioblastoma CSCs [115]. All of the mentioned functional steps of immune evasion and immunosuppression have also been described in medulloblastoma [116,117], but are not available in other types of paediatric CNS tumours.

Immune therapies in tumours of the CNS in children are challenged not only through the above immune evasion and immunosuppression processes, but also through the characteristics of the genomic landscape, which have revealed low microsatellite instabilities [118], as well as unique anatomical, physiological, and immunological barriers of CNS [119]. Still, there are aspirations, with the discoveries of the pan-expressed B7 homolog 3 protein (B7-H3) [120] and fat mass and obesity-associated protein (FTO) [121], which are powerful candidates for immunotherapeutic targets in paediatric CNS tumours in the future.

### 3.4. CSCs in Correlation with Cancer-Associated Fibroblasts

Cancer-associated fibroblasts (CAFs) are important components of the tumour microenvironment. There are several origins of CAF, such as the transference of fibroblasts in the host stroma, EMT, the transdifferentiation of perivascular cells, the differentiation of MSCs derived from bone marrow, etc. [83]. There has been evidence that CAF could also originate from CSCs during induced differentiation in other cancers [122], and the same behaviour in tumours of CNS has also attracted investigators since CAFs, or Glioma-associated stromal cells (GASCs), were identified with CAF functions of wound healing and angiogenesis [123]. However, most reports have suggested that these GASCs [124] originate from cells rather than CSCs, and recruited endothelial cells or reactive astrocytes with stem cell properties through EMT, transdifferentiated pericytes and vascular smooth muscle cells, and mesenchymal stem cells [124]. The latest research has found that glioma CSCs and GASCs cooperate with each other in protumoural effects, since CSCs chemotactically secrete PDGF and transforming growth factor beta (TGF-β) as mediators on GASCs, and GASCs induced CSC enrichment through osteopontin and HGF [125]. Tumour-associated pericyte (TAP) is another synonym of CAF in gliomas, and this fibroblast activation protein α (FAP) and the PDGFRβ positive cell can be targeted by oncolytic adenovirus [126].

There has been no report of CAF, GASC, or TAP from tumours of the CNS in children to date, yet their involvement in the reciprocation of CSCS and CAF maintenance and enrichment, the EMT of CSCs [127,128], and drug resistance [129] raise the importance of further investigating the features of these cells.

### 3.5. CSCs in Epithelial-Mesenchymal Transition

Metastasis is an eternal topic in basic research on and clinical practice in cancers. It is clear that EMT plays an indispensable role in this biological process. The phrase “migrating cancer stem cells” was introduced for cancer cell-combined EMT properties with a stem-cell-like phenotype, to explain the complex cancer cell-(EMT)-migrating CSC-(MET)-metastatic cancer cell reversion [81]. In an early report from brainstem glioma, the expression of β-catenin and E-/N-cadherin in patient samples revealed no obvious staining for E-cadherin, but higher β-catenin and N-cadherin levels in high-grade tumours, and worse outcomes indicated the EMT’s involvement in glioma aggressive behaviour [130]. In SHH medulloblastoma, the downregulation of miR-466f-3p, together with the concordant upregulation of VEGFa and Neuropilin 2, encouraged cell proliferation and the self-renewal ability of CSCs through the EMT process, which sustained the mesenchymal phenotype of SHH medulloblastoma CSCs [131]. In posterior fossa ependymoma, it was found that the mesenchymal-like tumour cells were transformed and associated with distal metastases through the activation of NFκB and AP-1 complexes, induced by TNF-α in combination with TGF-β1 [132], and enhanced EMT correlated with poor outcomes in children and adults with different types of ependymomas [133]. Notably, C19MC, the key alteration in ETMR, could suppress EMT-associated genes [134], which indicated ETMR with mutant C19MC might have enhanced the EMT profile. Lastly, EMT processes in high-grade glioma and pilocytic astrocytoma were simultaneously discovered, but higher EMT activities correlated with a higher grade of tumours and worse prognoses for the patients, while the existence of apparent aggressive glioblastoma subtypes with high E-cadherin expression [135,136] caused experts in the field to rethink EMT/MET in tumour aggressiveness.

### 3.6. CSCs in Multidrug Resistance

Chemoradiation resistance has plagued oncologists and cancer researchers for a long time. ATP binding cassette (ABC) transporters were the first family of proteins described in the function of transporting a range of substrates, including peptides, to conduct drug-resisting functions [137]. Clinical data has shown the ABCG2 transporter is increasingly expressed according to the tumour grades in gliomas, especially in CD133-positive CSCs. In an in vitro experiment, adding ABCG2 inhibitor nicardipine successfully enhanced the sensitivity to mitoxantrone in CSCs [138]. In ependymoma, the high expression level of ABCB1 was observed in the CSC population, and the inhibition of ABCB1 by vardenafil or verapamil could potentiate the response to chemotherapeutic drugs of vincristine, etoposide, and methotrexate [139]. The biological significance of ABC transporters expanded when the study on medulloblastoma showed the specificity of expression levels of different family members of ABC transporters, and that in SHH medulloblastoma, ABCA8 and ABCB4 levels were higher, while ABCC8 levels were lowest in the SHH subgroup [140], which may be due to the different origins of CSCs in different medulloblastoma subgroups. Although the function of CSC-expressed ABC transporters highlights that the binding of hydrophobic substrates, including chemotherapeutic drugs, leads to a conformation change, ATP binding, and the release of substrates outward the formation of a pore-like structure, the results of CSC-eliminating clinical trials indicate that molecules or antibodies targeting ABC transporters generally show a less convincing response rate in clinical settings [141]. This raises the importance of other participants in CSC multidrug resistance function.

The specific roles of CSC marker CD133 in CSC biology have been investigated in many kinds of cancers, as well as its involvement in the positive regulation of autophagy correlated with chemoresistance capability [142]. In low-grade glioma of pilocytic astrocytoma, the recurrent cases always face difficulty in readopt chemotherapies, while targeting CD133-positive CSCs can significantly improve chemotherapeutic efficacy [143]. Other CSC markers, such as SOX2 [144], Nanog [145], CD44 [146], etc., have been investigated, and their involvement in the chemoresistance of CSCs was confirmed through various signalling pathways.

The DNA damage repair process in CSCs is also a key area in the research field of chemoresistance. O6-alkylguanine repair, DNA mismatch repair, DNA base excision repair, and the sensing of DNA damage are all steps CSCs involve in the restoration of the chemoradiation-injured DNA. A large number of novel molecular drugs had been introduced in targeting CSCs’ function of chemoresistance [147].

## 4. Clinical Significance of Tumour Stem Cells in Tumours of the Central Nervous System in Children

Since CSCs are isolated in various types of CNS tumours in children and play critical roles in many biological processes in tumour development, their existence highlights the correlation of clinical practices regarding CNS tumours in children.

### 4.1. CSCs and Clinical Features

Although the hypothesis of the CSC model remains uncertain, either generated from normal stem cells, normal progenitor cells, or even dedifferentiated from mature tumour cells, the CSCs can perfectly unify hierarchical and stochastic models in carcinogenesis [98]. It has been revealed that CSCs are involved in determining tumour (sub)types, the timing of tumour developments, the positioning of tumour origins, and the inciting of tumour aggressiveness.

During the early exploration of the origins of CSCs in medulloblastoma, specialists noticed that alterations in the critical SHH signalling pathway for regulating proliferation on the granule neuron precursor could induce SHH subgroup medulloblastoma, which led researchers to consider the possibility that different medulloblastoma subtypes arose from CSCs with distinct cell lineages [148]. Recently, it has become clear that lineage-specific molecular signatures in medulloblastoma CSC played a decisive role in the formation of four molecular subgroups of medulloblastomas [149]. They were protein patched homolog 1 (PTCH1)-deleted CSCs with granule neuron progenitors or a neural stem cell lineage, WNT alterations in CSCs matched to the lower rhombic lip pontine mossy fibre lineage in the brainstem, CSCs with a mixed population of malignant cells with divergent differentiation along the cerebellar lineage, or CSCs closely associated with a neuronal differentiation metaprogram and the unipolar brush cell lineage arising from the upper rhombic lip.

The involvement of CSCs in the timing of tumour development relies on the process of acquired mutations in different origins of CSCs (normal stem cells, precursor cells, or progenitor cells). It is clear that the incidence of CNS tumours overall increases with age [150], due to long-term carcinogen exposure that may be mutagenic in oncogenes or tumour-suppressor genes in the origins of CSCs. This does not conflict with a high incidence of CNS malignancies in children because epigenetic regulation in paediatric CNS tumours is more apparent than in adult tumours, which are more influenced by environmental and microenvironmental features [151]. Genetic alterations in early embryogenic pathways in CSC origin cells and the time of gene alterations in different differentiation stages both contribute to the timing of tumour development, especially to distinct tumour types in early childhood. For example, the dysregulation of embryonal development pathways SHH, WNT, and NOTCH, or the alteration of the proto-oncogenic tumour suppressor Retinoblastoma protein (Rb) oncogene Rb, are commonly seen in paediatric embryonal CNS tumours and are rarely seen in adult CNS tumours; CSCs from embryonic stem cells, neural crest cells, and neural stem cells can form embryonal tumours, while those from less pluripotent neural stem cells and neuronal/glial progenitor cells mainly form malignant gliomas [80,152]. There has also been a hypothesis that the temporal identities of the cell of origin determine tumour malignancies, which emphasizes the importance of the early temporal window in the cell cycle, but not the stemness stages [153].

Ependymal tumours, on the other hand, were the first kind of tumours that elucidated that the anatomic locations of the tumours correlated with CSCs from different populations of progenitor cells in the tissues of origin. These different populations of ependymoma-CSCs (with a radial glia cell phenotype) had self-renewal abilities and were multipotent in culture; additionally, they also had anatomic site-specific chromosomal alterations or expression signatures, such as Ephrin type B receptor 2/3/4 (EPHB2/3/4), Ephrin A3/4, Jagged1/2 (JAG1/2) in supratentorial tumours, an inhibitor of a differentiation family of proteins 1/2/4 (ID1/2/4), AQP1/3/4 in anteroposterior tumours, and Homeobox (HOX) in spinal tumours [33], later confirmed by the single-cell RNA sequencing technique.

According to the results of single-cell RNA sequencing, the proportions of CSCs in ependymal tumours correlate with their aggressiveness, which enrich differentiated populations and CSCs with quiescent neural stem cell profiles, underlying their more indolent clinical behaviours in patients with YAP and PFB ependymomas, or tumours enriching differentiated lineages of PF-Ependymal-like and PF-Astroependymal-like programs in PFB and PF-subependymoma, which have been revealed to have distinctly favourable clinical outcomes; while tumours enriching undifferentiated PFA subpopulation are associated with more-aggressive clinical courses [35,36]. Intriguingly, in adult astrocytic tumours, we previously noticed that the frequency of Sox2/CD44v9-negative and phospho-S6-positive tumour cell populations (differentiated tumour cells), but not Sox2/CD44v9-positive and phospho-S6-negative CSC populations, correlated with the grade of glioma. This phenomenon revealed the complexity of research on CSCs by different methods or conditions, and pathways such as mTOR may contribute to tumour aggressiveness and the separate maintenance of stemness [154].

### 4.2. CSCs and Anti-Cancer Therapies

The first approach to treating tumours of the CNS in children is surgical resection, the extent of which is related to outcomes in children with low- or high-grade gliomas (except midline gliomas) [155,156]. Notably, evidence has increasingly shown that an operation might cause residual CSCs to survive and grow through postoperative angiogenesis, reactive astrogliosis, CSC niche formation, growth factor productions, and neuro-inflammation, due to surgical brain injury [157,158]. These clues come from glioblastoma in adults, where surgery-induced microglia/macrophage infiltration, angiogenesis, and the upregulation of stem-cell-related genes were confirmed by RNA sequencing in recurrent and primary tumour tissues [159]. Unfortunately, there are no reports on postoperative CSC status in paediatric CNS tumours yet, and maximal safe resection had limitations in eliminating all perilesional CSCs.

Chemotherapy is applied in many types of CNS tumours in children; however, the chemoresistance of CSCs has been as widely known by experts as early as the concept of CSCs was first accepted. Clinical evidence has shown the chemo-medication-induced CSC marker upregulated in breast cancer patients receiving neoadjuvant chemotherapies or an EGFR/HER2 inhibitor [160]. Since chemo-medications target vigorously proliferating cells, the quiescent CSCs can perfectly escape from chemotherapies through their intrinsic and extrinsic drug-resistant abilities, which are DNA sensor and repair pathways, the expression of drug transporters, EMT in the tumour microenvironment, and the shelter of the niche [161].

Radiation therapies are commonly introduced to combat malignant tumours such as paediatric high-grade gliomas, embryonal tumours, germ cell tumours, and benign tumours arising in special locations with functional impairment. A clinical study in breast cancers found that fractionated radiation not only spared CSCs, but also mobilized them from a quiescent phase into actively cycling cells, while the surviving non-tumourigenic cells were driven into senescence [162]. Similar to chemoresistance, the radioresistance of CSCs was mediated by redistribution in terms of the cell cycle, the enhanced repair of DNA, the scavenging of reactive oxygen species and free radicals, and the induction of inflammatory and tumourigenesis pathways in the CSC microenvironment [163]. There was also a hypothesis that irradiation could “awaken” quiescent CSCs to enter the cell cycle, causing tumour recurrence after initial treatments [164].

For over a decade, it has been known that the treatment principle of the elimination or induction of the differentiation of CSCs had become a promising way by which to deal with CSCs, through targeting pathways such as WNT, SHH, and NOTCH, which are required for the maintenance of CSCs [165]. The microenvironments of CSCs are also considerable targets, since they provide suitable space for self-renewal and increased CSC chemical and radiological tolerance, which include vascularization, hypoxia, tumour-associated macrophages/fibroblasts, and the extracellular matrix [83]. The drug and gene delivery systems, such as nanoparticles, have also been extensively studied for enhancing targeting therapies for CSCs in paediatric CNS tumours [166]. In recent preliminary reports, controversial results were achieved for the same (class) medications in several common paediatric CNS tumours by different study groups. In recurrent, progressed, or relapsed SHH group medulloblastoma, both SHH pathway inhibitors Vismodegib and Sonidegib, targeting transmembrane protein Smoothened (SMO), were revealed with anti-tumour activities in monotherapy stratums [167], while the combination of Vismodegib and Temozolomide failed as the proportion of successes required was not reached [168]. In trials of HDAC inhibitors Vorinostat and valproic acid, promising efficacies were achieved in relapsed/refractory neuroblastoma [169], and in very young children (less than 48 m/o) with medulloblastoma or ETMR [170], in a combination of other agents, while using continuous valproic acid with post-irradiation bevacizumab or monotherapy of Vorinostat, it failed to improve outcomes in children with DIPG [171,172]. In targeting immune checkpoints, great challenges were recognized since the tumour mutational burden (TMB) was relatively low in various CNS tumours in children [118]. In a large trial testing antibodies of Cytotoxic T-lymphocyte-associated protein 4 (CTLA 4) and programmed death-1 (PD-1), the monotherapy or combined therapy could not improve the survival rate in children with DIPG and advanced high-grade tumours of medulloblastoma, ependymoma, and other high-grade gliomas [173]. It is important to emphasize that the suitable target population should be carefully selected by elevated PD-L1 expression and high TMB, as a response of treatments is worlds apart [174].

Nevertheless, other novel therapeutic methods have also attracted experts to improve treatment efficacy. Chimeric antigen receptor T-cell (CAR-T) therapy has been considered as a candidate approach since the FDA has approved its application in haematological malignancies. The advances in the development of multi-antigen targeting CAR-T and the generation of CSC-specific CAR-T have enhanced confidence in its potential benefits in treating children with CNS tumours [175,176,177]. However, with no reports from large-scale trials and methodological challenges, such as the presence of the blood-brain barrier, the cross-expression of CSC markers in immature normal tissue, immune suppression by tumour microenvironment [178,179], treatment with CAR-T in targeting CSCs in paediatric CNS tumours still has a long way to go. The oncolytic virus was another potential way to target CSCs in children with CNS tumours, as an early review revealed multiple types of oncolytic viruses having the ability to kill CSCs [180]. A recent report on using oncolytic HSV-1 G207 in high-grade glioma in children showed satisfying safety data and preliminary positive antitumour efficacy [181]. Since there are a large number of ongoing studies on the oncolytic virus in paediatric CNS tumours, based on the results of trials in adults [182], evidence of this therapy is highly anticipated.

### 4.3. CSCs and Tumour Recurrences

Tumour relapse and metastasis are major causes of cancer-related mortality. There have been studies on paired primary and recurrent tumours using various techniques in determining the biological difference and novel targets for further treatments. It was revealed that cancers from different systems shared similar contributions of CSCs in tumour post-treatment recurrences, including the evidence of upregulated CSC markers, such as CD133 and CD44 variants in recurrent tissues, which might act as predictive, prognostic, and therapeutic targets [183,184,185,186]. In tumours of the CNS, early reports from adult high-grade gliomas revealed that the accumulation of CSCs in post-irradiated necrosis areas [187] or the expansion of CSCs in post-chemoradiation recurrences [188] and CSC markers were related to the time to recurrence and overall survival of these patients [189,190].

In children with CNS tumours, the detection of CSCs in clinical research has been rarely conducted, while basic research using RNA sequencing in both ependymoma and medulloblastoma has indicated a trend toward the enrichment of CSCs in recurrent tumours, surviving from multidisciplinary treatments [36,191]. The CSCs are involved in the rare phenomenon of radiation-induced glioblastoma (RIG) in children treated for medulloblastoma, since the RIG showed a similar CSC-specific profile with not only the primary medulloblastoma, but also with other astrocytomas and medulloblastomas [192].

We can rely on the great progress in clinical laboratory technology; tests on CSCs for diagnostic procedures or outcome predictions in children with CNS tumours may emerge in future practices. Liquid biopsy to obtain cell-free DNA from a patient’s blood or cerebrospinal fluid for methylation sequencing has provided reliable data on medulloblastoma [193]; this might be a candidate method in follow-up examinations for children with CNS tumours.

## 5. Conclusions

CSC is an enduring hotspot for medical workers and researchers in the oncology specialty. This comprehensive review considers the distinct CSCs in common types of tumours of the CNS in children by their CSC markers, discusses the biological functions in tumour initiation and maintenance, and describes their relationship with clinical practice. However, there are plenty of tasks awaiting further investigations in order to clarify the real model of CSCs, the real role of CSCs in tumourigenesis/progression, and a real therapy targeting CSCs. By increasing research and clinical trials on CSCs of the CNS tumours in children today, a better understanding of these CSCs may be achieved.

## Figures and Tables

**Figure 1 cancers-15-03154-f001:**
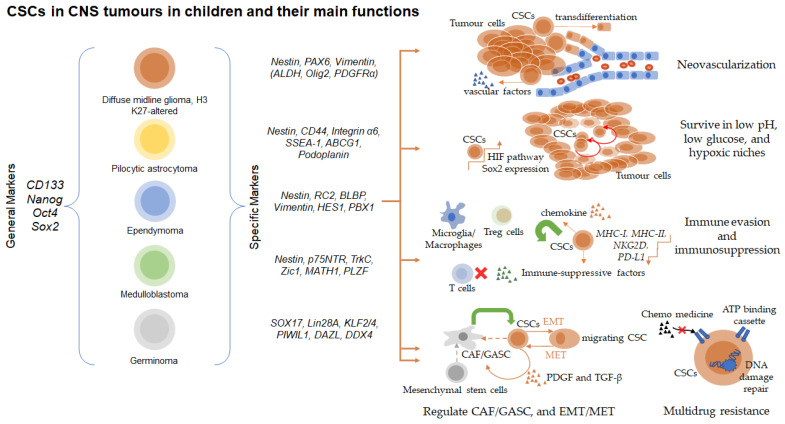
Common markers for CSCs of selected types of CNS tumours in children and their major biological functions.

**Table 1 cancers-15-03154-t001:** Tumour stem cell markers and markers for their differentiated forms from reports according to the WHO Classification of Tumours of the Central Nervous System 2021 edition.

Tumour Types	Markers for CSCs	Markers for Differentiated Cells	References
** *Gliomas, glioneuronal tumours, and neuronal tumours* **			
*Paediatric-type diffuse low-grade gliomas*			
Diffuse low-grade glioma, MAPK pathway-altered (previous oligodendroglioma)	CD133, SOX2, Nestin	CNPase, GFAP, NFP	[15]
*Paediatric-type diffuse high-grade gliomas*			
Diffuse midline glioma, H3 K27-altered	CD133, SOX2, Nestin, PAX6, Vimentin, (ALDH, Olig2, PDGFRα)	MAP2, GFAP, NeuN, NOTCH 1, CSPG4	[16,17,18,19]
Other types, except the abovementioned	Nestin, Musashi, CD133, SOX2, CD44, ALDH, L1CAM, Olig2, Nanog (partial, case by case), Podoplanin, Bmi-1, SEEA-1	Beta-III Tubulin, GFAP, O4, MAP2, NFP, CNPase, NeuN, MBP	[12,15,20,21,22]
*Circumscribed astrocytic gliomas*			
Pilocytic astrocytoma	CD133, Nestin, SOX2, Nanog, Oct4, CD44, Integrin α6 (CD49f), SSEA-1, ABCG1, Podoplanin, BLBP, A2B5, (Olig2, PDGFRα)	Beta-III Tubulin, GFAP, O4, S100B, NF, EAAT1/2, (Olig2, PDGFRα)	[11,23,24,25,26,27,28]
Pleomorphic xanthoastrocytoma	CD15	GFAP	[29]
*Glioneuronal and neuronal tumours*			
Dysembryoplastic neuroepithelial tumour	SOX2, AGR2	Beta-III Tubulin, GFAP	[30]
Central neurocytoma	CD133, Nestin, SOX2, Nanog, CD44, EGFR, hTERT, BLBP, GFAP-delta, Olig2, ASCL1	Beta-III Tubulin, GFAP, Synaptophysin, MAP2, NeuN, BMP2, BMPR1B, PSANACM	[31,32]
*Ependymal tumours*			
Supratentorial ependymoma	Nestin, CD133, RC2, BLBP, Vimentin, HES1, PBX1, SOX9, PAX3, Protein c-Fos, ZFP36, LGR5, EGR1, JUN, ATF3	Beta-III Tubulin, MAP2, GFAP, S100, CNPase, NG2, Synaptophysin, O4, Olig1/2, APC, CSPG2, DNAAF1, RSPH1, CAPS	[33,34,35,36]
Posterior fossa ependymoma			
Spinal ependymoma			
*Choroid Plexus Tumours*			
*Choroid plexus carcinoma*	MYC, Nestin, ATOH1, BLBP, GFAP, Geminin, GDF7	TTR, AQP1, OTX2, GMNC, MCIDAS, FOXJ1, CCNO, TAp73, MYB	[37,38,39]
*Embryonal Tumours*			
*Medulloblastoma*			
Medulloblastoma, SHH-activated	Nestin, Musashi, p75NTR, TrkC, Zic1, MATH1, SOX1, SOX2, PLZF, DACH1, Multimerin 1, PAX6, ATOH1, MycN	NeuN, GABRA6, Synaptophysin, PAX2, BLBP, O4	[12,40,41,42]
Medulloblastoma, non-WNT/non-SHH	CD133, Nestin, Musashi, SOX1, SOX2	Beta-III Tubulin, GFAP, CD44, CD24	[12,42]
*Other CNS embryonal tumours*			
Atypical teratoid/rhabdoid tumour	CD133, Nestin, Musashi, Nanog, Oct4, SOX2, ALDH, SALL4, MYC, LIN28A/B, NCAM, PAX6, KLF4	MAP2, Vimentin, GFAP, Synaptophysin, CD99, S-100, EMA, SMA	[43,44,45,46]
Embryonal tumour with multi-layered rosettes	LIN28A/B, HMGA2, Nestin, Vimentin, Oct4, SOX2, Nanog, CRABP1, DNMT3B, SOX3, SOX11, PAX6, SALL4, POU3F2, MEIS1/2, MYCN, Wee1, CHEK2	AQP4, GFAP, Synaptophysin, NeuN, NFP	[47,48,49]
*Germ cell tumours*			
*Germinoma*	SOX17, SOX2, Lin28A, KLF2/4, PIWIL1, DAZL, DDX4, NANOS3, ERVW-1, (Oct4, Nanog)	PLAP, KIT	[50]
*None germinoma germ cell tumours*	Lin28A, SOX2, KLF2/4, Oct4, Nanog	Not provided	[51]
** *Tumours of the sellar region* **			
*Adamantinomatous craniopharyngioma*	Nestin, SOX2, Oct4, CD133, KLF4, SOX9, β-catenin, MYC, SCA1, HESX1, KLF2/4	IT-1/POU1F1), TBX19/TPIT), SF-1/NR5A1	[52,53,54]
** *Pineal Tumours* **			
*Pineoblastoma*	CD133, Musashi, Podoplanin	Beta-III Tubulin	[55,56]
** *Cranial and Paraspinal Nerve Tumours* **			
*Schwannoma*	Oct4, SOX2, Nanog, MYC, KLF4, CD133, CD44, CXCR4	GFAP, S100, GAP43	[57]
*Neurofibroma*	PLP, Nestin, P75, GAP43, Sox10	GFAP, S100, GAP43	[58,59]
*Malignant peripheral nerve sheath tumour*			
** *Mesenchymal, non-meningothelial tumors* **			
*Uncertain differentiation (soft tissue tumours)*			
Ewing sarcoma	CD44, CD59, CD73, CD29, and CD54, CD90, CD105, CD166	SOX9, COL10A1, PPARg2, FABP4, LPL, SPP1, ALPL, RUNX2	[60]
** *Meningiomas* **			
*Meningioma*	Oct4, SOX2, Nanog, MYC, KLF4, CD133, Nestin	Not provided	[61]

Abbreviations of markers were listed below; markers controversially reported were listed in brackets. ABCG1: ATP Binding Cassette Subfamily G Member 1, AGR2: Anterior Gradient 2, ALDH: Aldehyde Dehydrogenase, ALPL: Alkaline Phosphatase, APC: Adenomatous Polyposis Coli, AQP1: Aquaporin 1, ASCL1: Achaete-Scute Homolog 1, ATF3: Activating Transcription Factor 3, ATOH1: Atonal BHLH Transcription Factor 1, BLBP: Brain Lipid Binding Protein, Bmi-1: B cell-specific Moloney Murine Leukaemia Virus Integration Site 1, BMP2: Bone Morphogenetic Protein 2, BMPR1B: Bone Morphogenetic Protein Receptor Type 1B, CAPS: Calcyphosine, CCNO: Cyclin O, CHEK2: Checkpoint Kinase 2, COL10A1: Collagen Type X Alpha 1 Chain, CRABP1: Cellular Retinoic Acid Binding Protein 1, CSPG: Chondroitin Sulfate Proteoglycan, CXCR4: C-X-C Chemokine Receptor Type 4, DACH1: Dachshund Family Transcription Factor 1, DAZL: Deleted in Azoospermia-like, DDX4: DEAD-Box Helicase 4, DNAAF1: Dynein Axonemal Assembly Factor 1, DNMT3B: DNA Methyltransferase 3 Beta, EAAT1/2: Excitatory Amino Acid Transporter 1/2, EGFR: Epidermal Growth Factor Receptor, EGR1: Early Growth Response Factor 1, EMA: Epithelial Membrane Antigen, ERVW-1: Endogenous Retrovirus Group W Member 1, FABP4: Fatty Acid Binding Protein, FOXJ1: Forkhead Box J1 GABRA6: Gamma-aminobutyric Acid Receptor Subunit Alpha-6, GAP43: Growth Associated Protein 43, GDF7: Growth differentiation factor 7, GFAP: Glial Fibrillary Acidic Protein, GMNC: Geminin Coiled-Coil Domain Containing, HES1: Hairy and Enhancer of Split-1, HESX1: Homeobox Expressed in ES Cells 1, HMGA2: High Mobility Group AT-Hook 2, hTERT: Human Telomerase Reverse Transcriptase, KIT: Tyrosine-protein Kinase, KLF4: Kruppel-like Factor 4, L1CAM: L1 Cell Adhesion Molecule, LGR5: Leucine-rich Repeat-containing G-protein Coupled Receptor 5, LIN28A/B: Lin-28 Homolog A/B, LPL: Lipoprotein lipase, MAP2: Microtubule-associated protein 2, MATH1: Mammalian Atonal Homolog 1, MBP: Myelin Basic Protein, MCIDAS: Multi-Ciliate Differentiation and DNA Synthesis Associated Cell Cycle Protein, MEIS1/2: Meis Homeobox 1/2, MYB: V-myb Avian Myeloblastosis Viral Oncogene Homolog, NANOS3: Nanos C2HC-Type Zinc Finger 3, NCAM: Neural Cell Adhesion Molecule, NF: Neurofibromin, NFP: Neurofilament Protein, NG2: Neuron-glial Antigen, NOTCH 1: Neurogenic Locus Notch Homolog Protein 1, OTX2: Orthodenticle homeobox 2, p75NTR: p75 Neurotrophin Receptor, PAX6: Paired Box 6, PBX1: Pre-B-cell Leukaemia Transcription Factor 1, PDGFRα: Platelet-Derived Growth Factor Receptor Alpha, PIT-1 or POU1F1: Pituitary-specific positive transcription factor 1, PIWIL1: Piwi Like RNA-Mediated Gene Silencing 1, PLAP: Placental Alkaline Phosphatase, PLP: Myelin Proteolipid Protein, PLZF: Promyelocytic Leukaemia Zinc Finger Protein, POU3F2: POU Class 3 Homeobox 2, PPARg2: Peroxisome Proliferator-Activated Receptor Gamma 2, PSANACM: Polysialic Acid-Neuronal Cell Adhesion Molecule, RC2: Radial Glial Cell Marker-2, RSPH1: Radial Spoke Head Component 1, RUNX2: Runt-related Transcription Factor 2, S100B: Calcium-Binding Protein B, SALL4: Spalt Like Transcription Factor 4, SCA1: Stem Cell Antigen 1, SF-1 or NR5A1: Steroidogenic Factor-1, SMA: Smooth Muscle Actin, SOX: Sex Determining Region Y-box, SPP1: Secreted Phosphoprotein 1, SSEA-1 or CD15: Stage-Specific Embryonic Antigen-1, TBX19 or TPIT: T-box Transcription Factor 19, TrkC: Tropomyosin Receptor Kinase C, TTR: Transthyretin, ZFP36: Zinc Finger Protein 36 Homolog, Zic1: Zinc Finger Protein of the Cerebellum 1.

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
