# Peer review of "Cancer Stem Cells in Tumours of the Central Nervous System in Children: A Comprehensive Review"

_cancers, 2023, doi:10.3390/cancers15123154_

Round 1

Reviewer 1 Report

Han et al., review recent advances in pediatric central nervous system cancer stem cells. This is a timely review as there are no recent reports on this topic. The content is comprehensive, reflecting a thorough review of recent advances.

2Table 1 was offset, thus I was only able to review the first 2 columns

3The tumor-specific molecular markers detailed in section 2 should be removed from the text as applicable if listed in table 1. Furthermore, the table can elaborate on these markers used for identification and/or putative therapeutic target.

 Section 3 could benefit from a Figure/illustration highlighting the multifaceted functions of CSC

1This diction of the manuscript can be greatly strengthened for clarity, reducing redundancy, better selection of word choices throughout the paper, and conveying the main points in a more succinct manner.  As an example, I revised section 1 below:

The increasing global incidence of CNS tumours [11, 12] is underscored by their poor prognosis [13]. Ongoing efforts to study paediatric CNS tumours culminated in the most recent WHO Classification of Tumours of the Central Nervous System focused on molecular subtypes [14]. However, these developments have translated to few novel treatments [15], necessitating development of mew therapeutic strategies.

Malignant tumours are comprised of diverse combinations of cells, with a relatively small subset of stem cell-like cancer cells possessing unlimited self-renewal potential [2, 3]. Cancer stem cells (CSCs) were first reported in leukemia, and were subsequently identified in other hematological malignancies and several solid tumours [1]. The heterogeneous nature of various malignancies, as well as phenotypical differences among patients with the same cancer type, mitigate efforts to identify, understand and develop targeted therapies against CSCs [4,5]. Following the seminal discovery of tumour stem-like cells from high grade gliomas [6], CSCs were subsequently identified in several paediatric CNS tumours: pilocytic astrocytoma, medulloblastoma, ganglioglioma, and ependymoma[7 -9]. Neurosphere assays are the standard procedure to isolate neural stem cells as well as CNS CSCs, suggesting CNS CSCs’s precursors are undifferentiated neural precursor cells [10]. We herein review recent scientific advances relating to pediatric CNS tumor CSCs and clinical implications.

Given the aforementioned comments, if the manuscript is substantially revised for diction and clarity, it would be amenable for acceptance. 

Reviewer 2 Report

The manuscript entitledCancer Stem Cells in Tumours of the Central Nervous System 2 in Children: A Comprehensive Reviewby Han and collaborators highlights the critical role that CSCs played in various types of CNS tumours in children as well as associated-biological processes for tumor initiation, recurrence and therapeutic resistance. The authors have done important work of documentation. This manuscript is well-written and can be of broad interest to those that investigate the development of tumours in children associated with CSCs. I have a few recommendations, which will be covered below:

1.      English needs some minor spell check.

2.      Table 1 must be on the same page. It’s very difficult to understand it.

3.    Consider including a graphical summary illustrating the cross-talk of CSCs among different CNS pediatric tumor types.

4.   Consider including a section on CSCs-targeted CRISPR-Cas9/CAR-T and therapy for pediatric CNS cancer treatment.

5.      Consider including a brief description of the current status of oncolytic virotherapy for pediatric CNS cancer treatment.

6.      Although the authors have included a perspective section, I suggest including a brief description on current challenges in treating pediatric brain tumors CSC-associated.

English needs some minor spell check.

Round 2

Reviewer 1 Report

Agree with all edits. Diction is significantly improved from initial draft.